# Convection Heat Transfer in 3D Wavy Direct Absorber Solar Collector Based on Two-Phase Nanofluid Approach

**Ammar I. Alsabery [1], Salma Parvin [2], Mohammad Ghalambaz [3,4], and Ali J. Chamkha [5,6] and Ishak Hashim [7,***

1   Refrigeration & Air-Conditioning Technical Engineering Department, College of Technical Engineering, The Islamic University, Najaf 54001, Iraq; alsabery_a@iunajaf.edu.iq
2   Department of Mathematics, Bangladesh University of Engineering and Technology, Dhaka 1000, Bangladesh; salpar@math.buet.ac.bd
3   Metamaterials for Mechanical, Biomechanical and Multiphysical Applications Research Group, Ton Duc Thang University, Ho Chi Minh City 758307, Vietnam; mohammad.ghalambaz@tdtu.edu.vn
4   Faculty of Applied Sciences, Ton Duc Thang University, Ho Chi Minh City 758307, Vietnam
5   Faculty of Engineering, Kuwait College of Science and Technology, Doha District 35001, Kuwait; a.chamkha@kcst.edu.kw
6   Center of Excellence in Desalination Technology, King Abdulaziz University, P.O. Box 80200, Jeddah 21589, Saudi Arabia
7   Department of Mathematical Sciences, Faculty of Science & Technology, Universiti Kebangsaan Malaysia, UKM Bangi 43600, Selangor, Malaysia
*   Correspondence: ishak_h@ukm.edu.my; Tel.: +603-8921-5758

**Abstract:** A numerical attempt of the two-phase (non-homogeneous) nanofluid approach towards the convection heat transfer within a 3D wavy direct absorber solar collector is reported. The solar collector is permeated by a water-$Al_2O_3$ nanofluid and contains a wavy glass top surface that is exposed to the ambient atmosphere and a flat steel bottom surface. The left and right surfaces are maintained adiabatic. The governing equations of the Navier–Stokes and energy equations for the nanofluid are transformed into a dimensionless pattern and then solved numerically using the Galerkin weighted residual finite-element technique. Validations with experimental and numerical data are performed to check the validity of the current code. Impacts of various parameters such as the number of oscillations, wave amplitude, Rayleigh number and the nanoparticles volume fraction on the streamlines, isotherms, nanoparticle distribution, and heat transfer are described. It is found that an augmentation of the wave amplitude enhances the thermophoresis and Brownian influences which force the nanoparticles concentration to display a nonuniform trend within the examined region. Furthermore, the heat transfer rate rises midst the growing wave amplitude and number of oscillations. More importantly, such enhancement is observed more significantly with the variation of the wave amplitude.

**Keywords:** convection heat transfer; thermophoresis and Brownian; 3D wavy solar collector; two-phase nanofluid approach; finite element method

## 1. Introduction

Nowadays, the energy sector plays an critical role in any functional society [1]. The space between demand and supply expands due to the unprecedented increase in energy demand [2], which leads to a continuous consumption of fossil fuel. Furthermore, excessive utilization of fossil fuels causes

environmental pollution and global warming [3]. All the factors, as mentioned above, have drawn attention to researchers all over the world to use renewable energy technologies [4], such as solar energy [5,6]. Solar energy may be a solution to the current energy crisis, as the amount of energy in the solar flux landing on the Earth's surface in an hour is greater than all the energy consumed by humans in a year [7]. The difficulty extends toward collecting this energy efficiently and transforming it into something valuable. A solar thermal collector is a device which utilizes the solar energy via collecting and concentrating solar radiation [8,9]. The collected thermal energy is transferred through a flowing fluid that can be used in various thermal applications, such as water heating, building heating, and other industrial applications [10]. Nevertheless, solar energy has a huge operation expense and low-efficiency [11], which motivates many researchers to optimize and enhance the performance of such devices [12]. In the case of a standard model, particularly concerning a solar thermal collector that employs a black cover as the absorber which later conveys heat toward the fluid, the efficiency by which the absorber gains and transfers heat into the working fluid remains inadequate. The direct absorption solar collector (DASC) is a current production of solar collectors into which solar radiation is directly absorbed via a transport medium [13]. In the 1970s, DASC was initially introduced for improving the efficiency of collectors; the system incorporates solar flux in the employed fluid [14–16] directly.

Typical thermal liquids mentioned before such as water, ethylene/propylene glycol, and oil perform a vital purpose in numerous engineering sectors, including chemical production, electronic applications, power generation, heating and cooling processes, air conditioning, microelectronics, space and defense, and nuclear reactor cooling. These conventional fluids have weak thermal properties than the solids [17]. Various attempts for enhancing the heat transfer production of these liquids such as microchannels and fins with extended-surface, injection/suction of fluids, electrical/magnetic fields, and the vibration have been made; however, all these attempts have reached a dead end. New technologies such as nanotechnology and nanoscience utilizing the improved thermo-physical properties of conventional fluids have become a vital issue in heat transfer science [18,19].

An efficiency improvement of solar thermal collectors via employing nanofluids using DASC was demonstrated in various researches [20–24]. A comparison concerning the single- and two-phase procedures of nanofluids has been examined in several investigations [25–27]. A few authors also studied the three-dimensional behavior of nanofluids [28,29]. Surface waviness is a very efficient way to enhance heat transfer in solar collecting devices [30–32].

There is a significant number of achievements within the scope of heat transfer mode throughout direct absorption solar collectors; however, the 3-D model regarding the enhancement of collector efficiency, introducing surface waviness and nanofluid for such devices is yet to be studied. In this study, we focus on the two-phase nanofluid approach in a 3D wavy direct absorber solar collector. Moreover, the influences of various governing parameters as well as physical parameters are also discussed.

## 2. Mathematical Formulation

Consider the steady convective heat transfer phenomenon toward a 3D wavy direct absorber solar collector with length *L*, width *W*, and thickness *H* as reported in Figure 1. The solar collector contains a wavy glass top surface that is exposed to the ambient atmosphere and a flat steel (dark-colored) bottom surface. The left and right vertical surfaces are maintained adiabatic. The edges from the working region remain impermeable, and the area within the wavy solar collector surfaces remains filled with a water-$Al_2O_3$ nanofluid. The Boussinesq approximation holds true. Concerning the assumptions mentioned earlier, the Navier–Stokes and energy equations of the Newtonian fluid (nanofluid flow is incompressible and laminar) are addressed as [33,34]

$$\nabla \cdot \mathbf{v} = 0, \tag{1}$$

$$\rho_{nf}\mathbf{v} \cdot \nabla\mathbf{v} = -\nabla p + \nabla \cdot (\mu_{nf}\nabla\mathbf{v}) + (\rho\beta)_{nf}(T - T_c)\vec{g}, \tag{2}$$

$$(\rho C_p)_{nf}\mathbf{v} \cdot \nabla T = -\nabla \cdot (k_{nf}\nabla T) - C_{p,p}J_p \cdot \nabla T, \tag{3}$$

$$\mathbf{v} \cdot \nabla\varphi = -\frac{1}{\rho_p}\nabla \cdot J_p. \tag{4}$$

Here, $\mathbf{v}$ holds the 3D velocity vector, $\vec{g}$ denotes the vector of gravitational acceleration, $\varphi$ signifies the local volume fraction concerning nanoparticles, and $J_p$ implies the nanoparticles mass flux. According to the two-phase nanofluid approach, the nanoparticles mass flux is formulated as [33]

$$J_p = J_{p,B} + J_{p,T}, \tag{5}$$

$$J_{p,B} = -\rho_p D_B \nabla\varphi, \quad D_B = \frac{k_b T}{3\pi\mu_f d_p}, \tag{6}$$

$$J_{p,T} = -\rho_p D_T \frac{\nabla T}{T}, \quad D_T = 0.26\frac{k_f}{2k_f + k_p}\frac{\mu_f}{\rho_f T}\varphi. \tag{7}$$

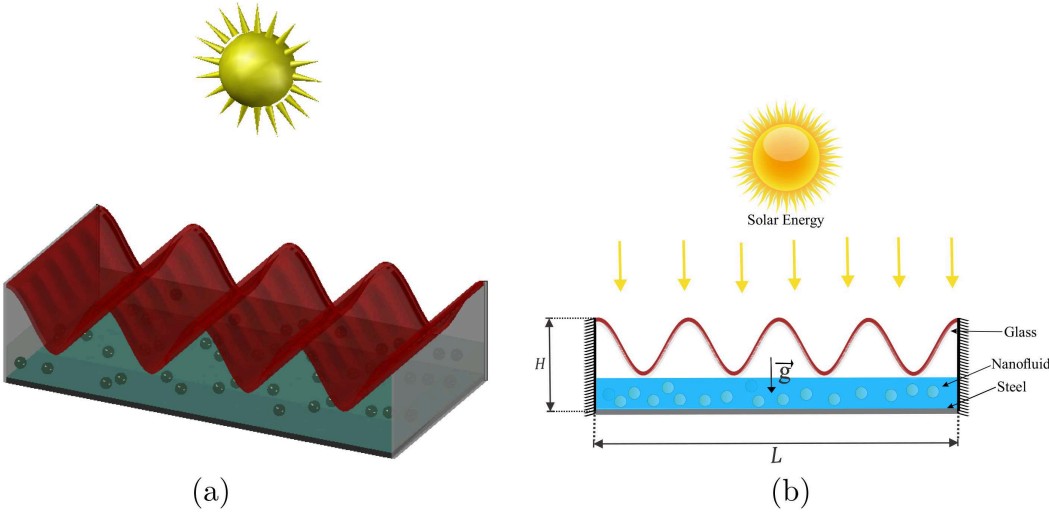

(a)               (b)

**Figure 1.** (**a**) 3D-Schematic diagram of the physical model and (**b**) 2D-Schematic diagram in the plane $(x, y)$.

The following can describe the applied thermophysical characteristics of $Al_2O_3$-water nanofluid [35,36].

$$(\rho C_p)_{nf} = (1 - \varphi)(\rho C_p)_f + \varphi(\rho C_p)_p, \tag{8}$$

$$\alpha_{nf} = \frac{k_{nf}}{(\rho C_p)_{nf}}, \tag{9}$$

$$\rho_{nf} = (1 - \varphi)\rho_f + \varphi\rho_p, \tag{10}$$

$$(\rho\beta)_{nf} = (1 - \varphi)(\rho\beta)_f + \varphi(\rho\beta)_p. \tag{11}$$

While the dynamic viscosity ratio of nanofluid for 33 nm particle-size is calculated as Corcione [35]

$$\frac{\mu_{nf}}{\mu_f} = \frac{1}{1 - 34.87\left(\frac{d_p}{d_f}\right)^{-0.3}\varphi^{1.03}}, \tag{12}$$

and the thermal conductivity ratio as Corcione [35]

$$\frac{k_{nf}}{k_f} = 1 + 4.4 \text{Re}_B^{0.4} \text{Pr}^{0.66} \left(\frac{T}{T_{fr}}\right)^{10} \left(\frac{k_p}{k_f}\right)^{0.03} \varphi^{0.66},$$ (13)

where $\text{Re}_B$ is evaluated as

$$\text{Re}_B = \frac{\rho_f u_B d_p}{\mu_f}, \quad u_B = \frac{2k_b T}{\pi \mu_f d_p^2},$$ (14)

where $k_b = 1.380648 \times 10^{-23} (J/K)$ is the Boltzmann constant. $l_f = 0.17$ nm is the mean path of fluid particles. $d_f$ is the molecular diameter of water given as [35]

$$d_f = 0.1 \left[\frac{6M}{N^* \pi \rho_f}\right]^{\frac{1}{3}}.$$ (15)

By presenting the following non-dimensional variables,

$$X = \frac{x}{L}, \quad Y = \frac{y}{L}, \quad Z = \frac{z}{L}, \quad \mathbf{V} = \frac{\mathbf{v}L}{\nu_f}, \quad \theta = \frac{T - T_c}{T_h - T_c},$$

$$\varphi^* = \frac{\varphi}{\phi}, \quad \text{Pr} = \frac{\nu_f}{\alpha_f}, \quad Ra = \frac{g\beta_f (T_h - T_c) L^3}{\nu_f \alpha_f}, \quad P = \frac{pL^2}{\rho_f \alpha_f^2}.$$ (16)

Equations (1)–(4), by using Equations (5)–(16), now become

$$\nabla \cdot \mathbf{V} = 0,$$ (17)

$$\mathbf{V} \cdot \nabla \mathbf{V} = -\nabla P + \frac{\rho_f}{\rho_{nf}} \frac{\mu_{nf}}{\mu_f} \nabla^2 \mathbf{V} + \frac{(\rho\beta)_{nf}}{\rho_{nf}\beta_f} \frac{1}{\text{Pr}} Ra \, \theta,$$ (18)

$$\mathbf{V} \cdot \nabla \theta = \frac{(\rho C_p)_f}{(\rho C_p)_{nf}} \frac{k_{nf}}{k_f} \frac{1}{\text{Pr}} \nabla^2 \theta + \frac{(\rho C_p)_f}{(\rho C_p)_{nf}} \frac{D_B^*}{\text{Pr} \cdot Le} \nabla \varphi^* \cdot \nabla \theta$$

$$+ \frac{(\rho C_p)_f}{(\rho C_p)_{nf}} \frac{D_T^*}{\text{Pr} \cdot Le \cdot N_{BT}} \frac{\nabla \theta \cdot \nabla \theta}{1 + \delta\theta},$$ (19)

$$\mathbf{V} \cdot \nabla \varphi^* = \frac{D_B^*}{Sc} \nabla^2 \varphi^* + \frac{D_T^*}{Sc \cdot N_{BT}} \cdot \frac{\nabla^2 \theta}{1 + \delta\theta},$$ (20)

where $\mathbf{V}$ is the dimensionless velocity vector $(U, V, W)$.

The dimensionless boundary conditions considering Equations (17)–(20) are provided by

On the top wavy surface:

$$\mathbf{V} = 0, \quad \frac{\partial \varphi^*}{\partial n} = -\frac{D_T^*}{D_B^*} \cdot \frac{1}{N_{BT}} \cdot \frac{1}{1 + \delta\theta_{nf}} \frac{\partial \theta_{nf}}{\partial n}, \quad \theta = 1,$$ (21)

On the bottom flat surface:

$$\mathbf{V} = 0, \quad \frac{\partial \varphi^*}{\partial n} = -\frac{D_T^*}{D_B^*} \cdot \frac{1}{N_{BT}} \cdot \frac{1}{1 + \delta\theta_{nf}} \frac{\partial \theta_{nf}}{\partial n}, \quad \theta = 0,$$ (22)

On the adiabatic right vertical surafce:

$$\mathbf{V} = 0, \quad \frac{\partial \varphi^*}{\partial n} = 0, \quad \frac{\partial \theta}{\partial n} = 0,$$ (23)

On the adiabatic left vertical surafce:

$$\mathbf{V} = 0, \quad \frac{\partial \varphi^*}{\partial n} = 0, \quad \frac{\partial \theta}{\partial n} = 0.$$ (24)

The local heat transfer (Nusselt number) is determined with the exposed top wavy surface from the following:

$$Nu_{nf} = -\frac{k_{nf}}{k_f} \sqrt{\left(\frac{\partial \theta}{\partial X}\right)^2 + \left(\frac{\partial \theta}{\partial Y}\right)^2 + \left(\frac{\partial \theta}{\partial Z}\right)^2}, \tag{25}$$

and the average Nusselt number ($\overline{Nu}_{nf}$) is determined via integrating the local heat transfer adjacent to the top wavy surface:

$$\overline{Nu}_{nf} = \frac{1}{S} \int_0^S Nu \, dS. \tag{26}$$

## 3. Numerical Method and Validation

The dimensionless governing equations Equations (17)–(20) including the selected boundary conditions Equations (21)–(24) are determined via the Galerkin weighted residual finite-element technique. Let $\nabla = \left(\frac{\partial}{\partial X}, \frac{\partial}{\partial Y}, \frac{\partial}{\partial Z}\right)$, and $\mathbf{V} = (U, V, W)$, the finite element analysis concerning the momentum equation Equation (18) in vectorial form is shown by the following steps.

The weak formulation regarding Equation (18) through multiplying this equation with an internal region of ($\Phi$) and integrating that across the computational area which remains discretized toward small triangular components as revealed within Figure 2. The resulting weak formulation is reached as

$$\int_\Omega \Phi_i \mathbf{V}^k \left(\frac{\partial \mathbf{V}^k}{\partial X} + \frac{\partial \mathbf{V}^k}{\partial Y} + \frac{\partial \mathbf{V}^k}{\partial Z}\right) dXdYdZ = -\int_\Omega \Phi_i P^k \left(\frac{\partial}{\partial X} + \frac{\partial}{\partial Y} + \frac{\partial}{\partial Z}\right) dXdYdZ$$

$$+ \frac{\rho_f}{\rho_{nf}} \frac{\mu_{nf}}{\mu_f} \int_\Omega \Phi_i \mathbf{V}^k \left(\frac{\partial^2}{\partial X^2} + \frac{\partial^2}{\partial Y^2} + \frac{\partial^2}{\partial Z^2}\right) dXdYdZ + \frac{(\rho\beta)_{nf}}{\rho_{nf}\beta_f} \frac{Ra}{Pr} \int_\Omega \Phi_i \theta^k dXdYdZ.$$

Selection of the interpolation functions for providing an approximation for the velocity distribution, pressure, and temperature distribution as

$$\mathbf{V} \approx \sum_{j=1}^m \mathbf{V}_j \Phi_j(X, Y, Z), \quad P \approx \sum_{j=1}^m P_j \Phi_j(X, Y, Z), \quad \theta \approx \sum_{j=1}^m \theta_j \Phi_j(X, Y, Z).$$

The nonlinear residual equation for Equation (18) that is obtained from the Galerkin weighted residual finite-element method is

$$R_i = \sum_{j=1}^m \mathbf{V}_j \int_\Omega \left[\left(\sum_{j=1}^m \mathbf{V}_j \Phi_j\right) \frac{\partial \Phi_j}{\partial X} + \left(\sum_{j=1}^m \mathbf{V}_j \Phi_j\right) \frac{\partial \Phi_j}{\partial Y} + \left(\sum_{j=1}^m \mathbf{V}_j \Phi_j\right) \frac{\partial \Phi_j}{\partial Z}\right] \Phi_i dXdYdZ$$

$$+ \sum_{j=1}^m P_j \left(\int_\Omega \frac{\partial \Phi_j}{\partial X} + \int_\Omega \frac{\partial \Phi_j}{\partial Y} + \int_\Omega \frac{\partial \Phi_j}{\partial Z}\right) \Phi_i dXdYdZ$$

$$+ \frac{\rho_f}{\rho_{nf}} \frac{\mu_{nf}}{\mu_f} \sum_{j=1}^m \mathbf{V}_j \int_\Omega \left[\frac{\partial \Phi_i}{\partial X} \frac{\partial \Phi_j}{\partial X} + \frac{\partial \Phi_i}{\partial Y} \frac{\partial \Phi_j}{\partial Y} + \frac{\partial \Phi_i}{\partial Z} \frac{\partial \Phi_j}{\partial Z}\right] dXdYdZ$$

$$+ \frac{(\rho\beta)_{nf}}{\rho_{nf}\beta_f} \frac{Ra}{Pr} \int_\Omega \left(\sum_{j=1}^m \theta_j \Phi_i\right) \Phi_i dXdYdZ,$$

where the superscripts *k*, *i*, *j*, and *m* are approximate index, residual number, node number, and iteration number, respectively. For the intention of simplifying these nonlinear expressions into the momentum equations, any Newton–Raphson iteration algorithm is employed. The current

convergence of the solution exists when the relative error toward variables remains according to the following convergence criteria,

$$\left| \frac{\Gamma^{m+1} - \Gamma^m}{\Gamma^{m+1}} \right| \leq 10^{-5}.$$

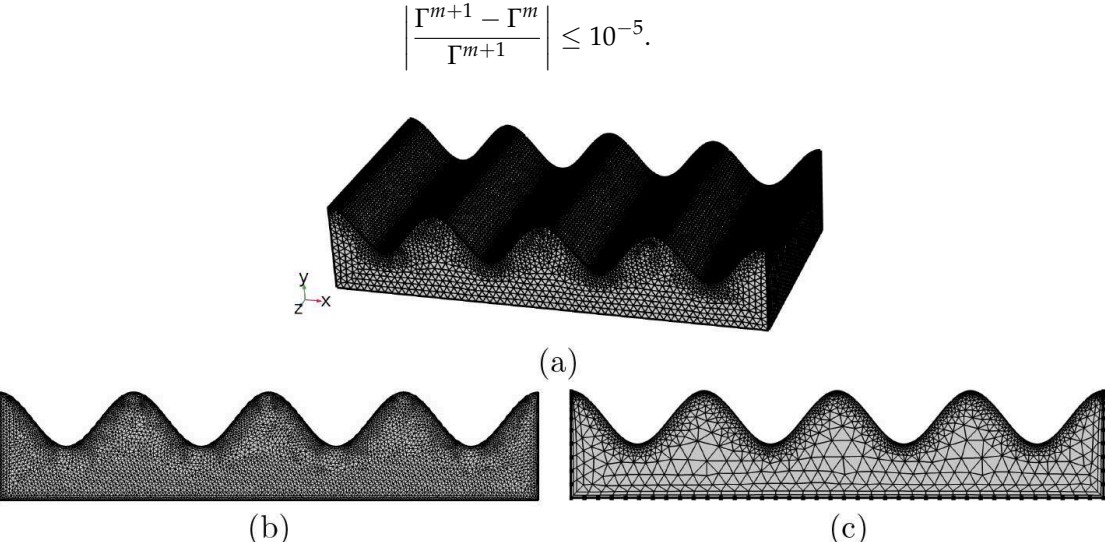

**Figure 2.** (**a**) 3D grid-points distribution for grid size of 364,856 elements, (**b**) 2D grid-points distribution for grid size of 9463 elements and (**c**) 2D grid-points distribution for grid size of 2367 elements in the plane $(X, Y)$.

Toward the aim of approving the existing numerical data, the outcomes of the current study are compared to the outcomes described by Corcione et al. [37] concerning the mode of free convection inside a square hollow cavity of heated of sides, as designated in Figure 3. Besides, Figure 4a shows a comparison between the current outcomes and the experimental arrangements of Putra et al. [38] and the numerical outcomes of Corcione et al. [37] using the two-phase nanofluid model for various Rayleigh numbers at $\phi = 0.01$ and $N = 0$. While for the adopted nanofluid models of the thermal conductivity and dynamic viscosity, a validation is achieved with other experimental/numerical data, as given in Figure 4b,c. Those outcomes produce a clear confidence in the accuracy of the existing numerical approach.

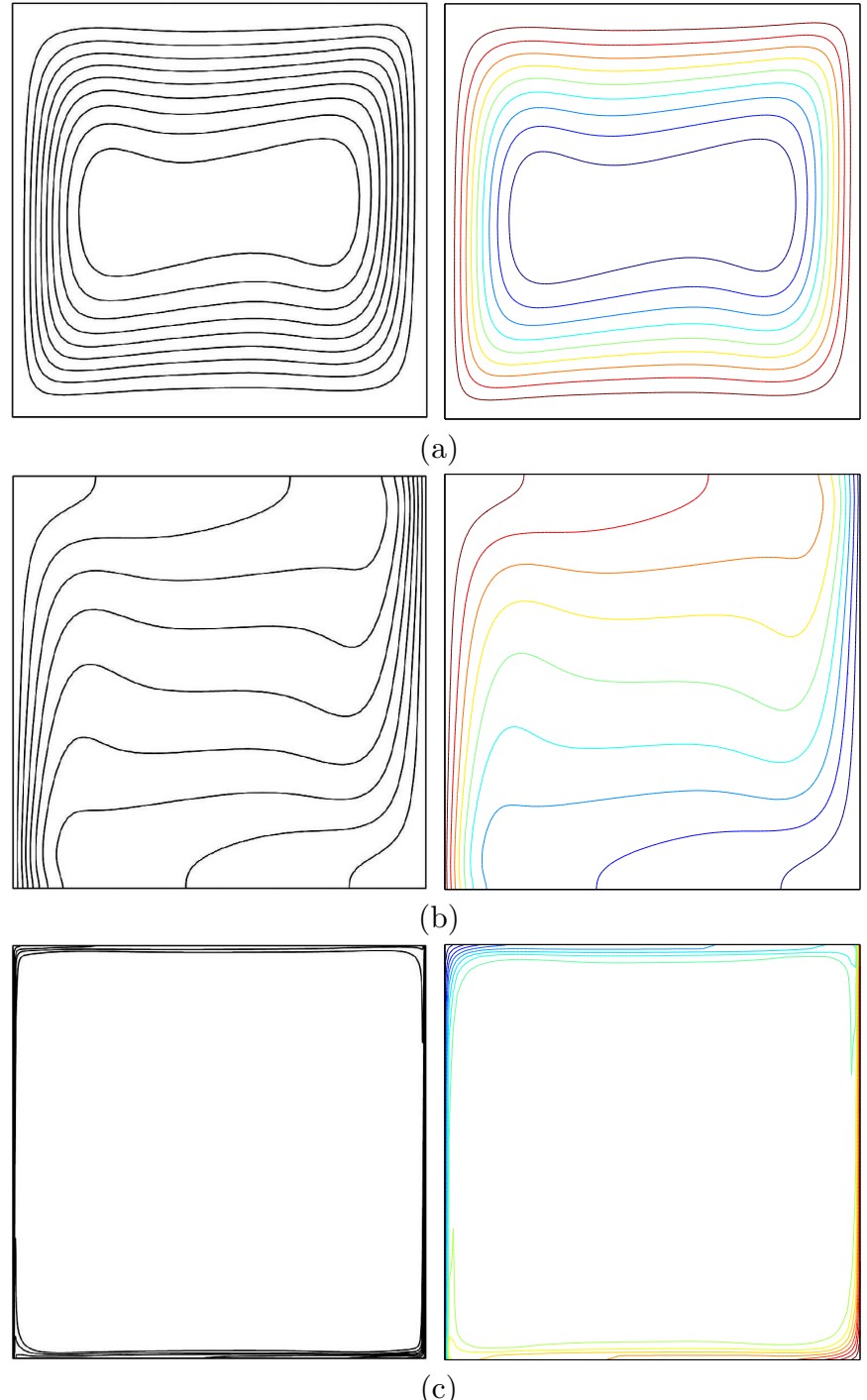

**Figure 3.** (left) Corcione et al. [37] and (right) current work for (**a**) streamlines, (**b**) isotherms and (**c**) nanoparticle distribution at $Ra = 3.37 \times 10^5$, $N = 0$, $\phi = 0.04$, $H = 1$ and $D = 0$.

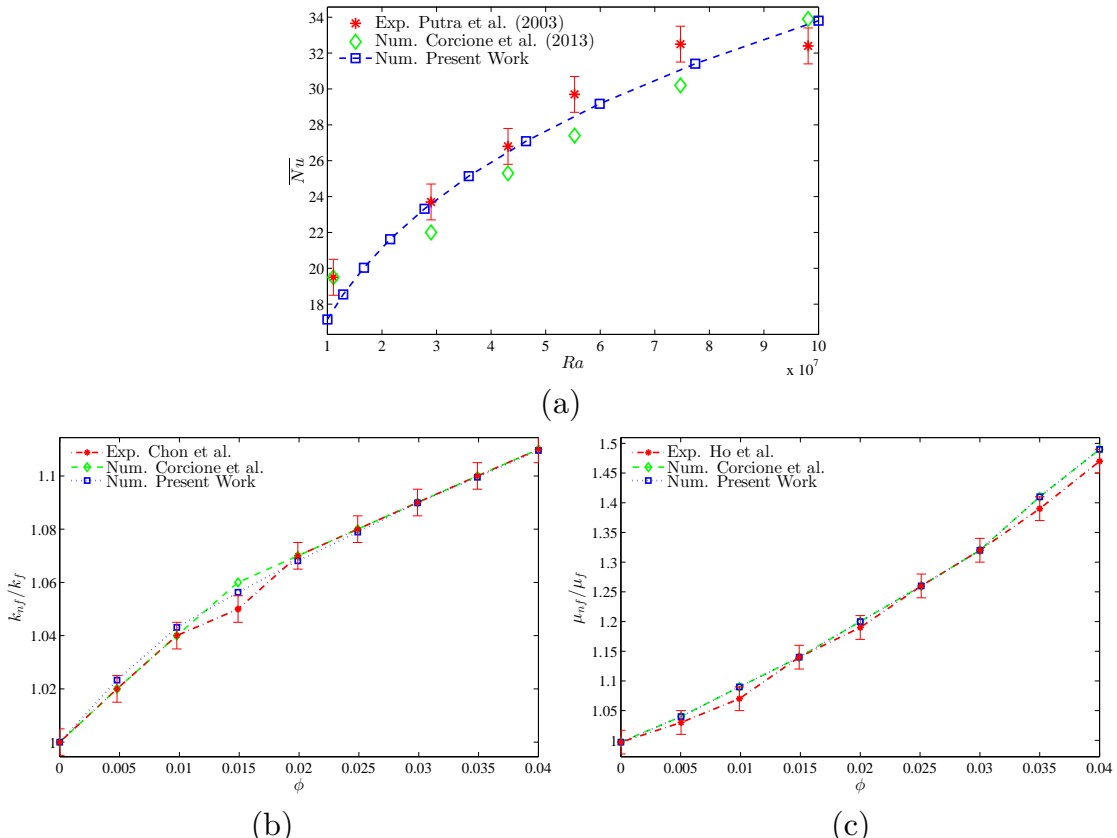

**Figure 4.** Validations of (**a**) average Nusselt number of the current numerical work with the experimental outcomes of Putra et al. [38] and the numerical outcomes of Corcione et al. [37] for various Rayleigh numbers at $\phi = 0.01$ and $N = 0$, (**b**) thermal conductivity ratio with experimental outcomes of Chon et al. [39] and numerical data of Corcione et al. [37], and (**c**) dynamic viscosity ratio with experimental outcomes of Ho et al. [40] and numerical data of Corcione et al. [37].

## 4. Results and Discussion

This section presents numerical results for the streamlines, isotherms, and nanoparticle distribution for five parameters. These are number of oscillations ($1 \leq N \leq 4$), amplitude ($0.05 \leq A \leq 0.15$), Rayleigh number ($10^4 \leq Ra \leq 10^7$), and nanoparticle volume fraction ($0 \leq \phi \leq 0.04$). The contents concerning the remaining parameters remain fixed at $\mathrm{Pr} = 4.623$, $Le = 3.5 \times 10^5$, $Sc = 3.55 \times 10^4$, $N_{BT} = 1.1$, and $\delta = 155$. Notably, the conservation equations remain entirely coupled. Such **V** depends toward $\phi$ through viscosity; $\phi$ relies upon $\theta$ mainly due to thermophoresis force; $\theta$ relies on $\phi$ by thermal conductivity including the Brownian and thermophoretic effects inside the energy equation; and $\phi$ and $\theta$ depend toward **V** due to the convection phases within the nanoparticle momentum and energy equations, respectively. The employed thermo-physical features of the working liquid (water) and the solid $Al_2O_3$ phases are listed in Table 1.

**Table 1.** Thermo-physical properties of water with $Al_2O_3$ nanoparticles at $T = 310$ K [41].

| Physical Properties | Fluid Phase (Water) | $Al_2O_3$ |
|---|---|---|
| $k\,(\mathrm{Wm^{-1}K^{-1}})$ | 0.628 | 40 |
| $\mu \times 10^6\,(\mathrm{kg/ms})$ | 695 | – |
| $\rho\,(\mathrm{kg/m^3})$ | 993 | 3970 |
| $C_p\,(\mathrm{J/kgK})$ | 4178 | 765 |
| $\beta \times 10^5\,(1/\mathrm{K})$ | 36.2 | 0.85 |
| $d_p\,(\mathrm{nm})$ | 0.385 | 33 |

## 4.1. Outcomes of Number of Oscillations (N)

The effects of the number of oscillations (*N*) on the streamline, isotherm, and nanoparticle distribution for 3-D and in the $X − Y$ plane are illustrated in Figures 5 and 6, respectively. In this case, the wave amplitude (*A*), Rayleigh number (*Ra*), and nanoparticle volume fraction ($\phi$) are chosen as 0.1, $10^6$, and 0.02, respectively. The flow, thermal, and concentration fields are more clearly visible in the cross-sectional plots as shown in Figure 6. The flow field is driven by the buoyancy force which consists of two symmetrical vortices near each wave as seen from the first columns of Figures 5 and 6. For every wave, the vortices are of equal and opposite strength. It is observed that, due to the increasing number of oscillations, the vortices become weaker. The value of stream function falls to 1 from 4 for raising the oscillations from 1 to 5. This is due to the fact that for fixed length of the bottom surface of the collector, increasing waviness of the upper surface the wave length reduces that leads the reduction of flow circulation.

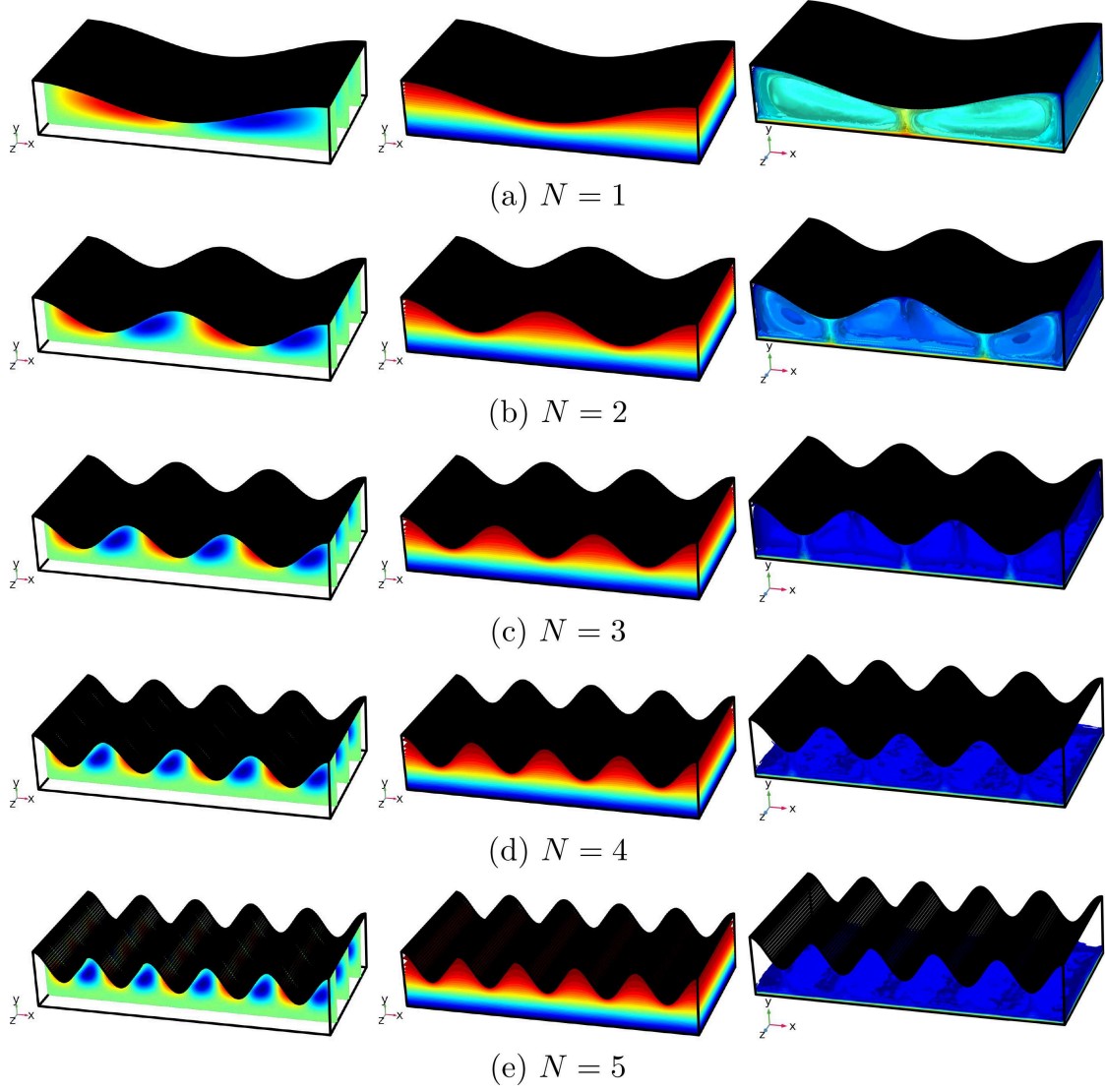

(a) $N = 1$

(b) $N = 2$

(c) $N = 3$

(d) $N = 4$

(e) $N = 5$

**Figure 5.** Variation of the 3D streamlines (**left**), isotherms (**middle**), and nanoparticle distribution (**right**) evolution by number of oscillations (*N*) for $A = 0.1$, $Ra = 10^6$, and $\phi = 0.02$.

The middle columns of Figures 5 and 6 present the isotherm contours for various values of the *N*. Here, the isotherms inside the collector are governed mainly by the convection due to the strong buoyancy effect. The isotherms are parallel to the bottom wall and take wavy shapes near the top wavy surface. For $N = 1$, the higher temperature gradients are restricted near the concave

portion of the upper wall, which also indicates a region of higher heat transfer because the distance between the upper and lower surface is minimum at this area. For increasing the value of $N$ to 5, the higher temperature gradient region is extended to the five concave portions of the top wall. For this reason, extra energy is conveyed from the upper wall to the bottom wall making the inside fluid hotter compared with other considered lower number of oscillations. The nanoparticle distributions for different $N$ are presented in the last columns of Figures 5 and 6. The particle diffusion is governed by the Brownian motion and thermophoresis effect in addition to the convection current. Most of the iso-concentration lines appear near the collector walls and take the wall pattern. It is clearly seen from the figures, that at $N = 1$, higher and lower concentration gradients remain at the convex and concave regions, respectively. This trend continues up to the value of $N = 3$. However, further increment of $N$ leads the different pattern of concentration gradient at the bottom wall. Though the same concentration was applied to the upper and lower boundary, increasing waviness causes the upper wall to have a larger surface area. This is why lower concentration lines appear at the bottom surface for higher values of $N$.

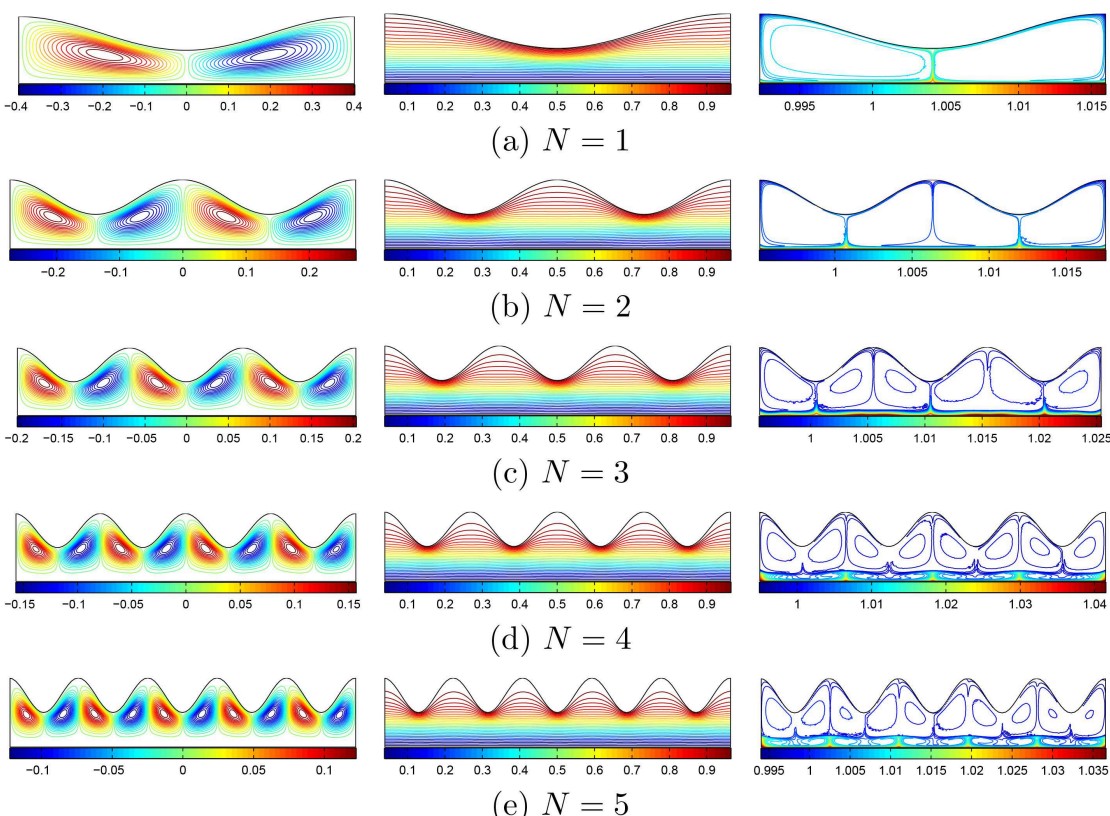

**Figure 6.** Variation of the streamlines (**left**), isotherms (**middle**), and nanoparticle distribution (**right**) evolution by number of oscillations ($N$) for $A = 0.1$, $Ra = 10^6$ and $\phi = 0.02$ in the plane $(X, Y)$.

### 4.2. Outcomes of Wave Amplitude (A)

The 3-D plots and the cross-sectional view in the $X - Y$ plane of the streamlines, isotherms, and the nanoparticle distribution for various wave amplitudes ($A$) are depicted in Figures 7 and 8, respectively. The numbers of waves ($N$), Rayleigh number ($Ra$) and nanoparticle volume fraction ($\phi$) are kept fixed at 4, 106, and 0.02, respectively. From the first columns of Figures 7 and 8, it can be seen that the stream lines consist of four pairs of strong recirculation cells at the upper part occupying most part of the domain and eight comparatively lower strength cell at the lower part of the domain for $A = 0.05$. For increasing wave length the vortices at the lower part become weaker and finally disappear at $A = 0.125$, on the other hand, the vortices at the upper portion become larger in size and strength and occupy the whole domain as the values of $A$ rises. On the increase of wave amplitude, wavelength

increases which leads the strong circulation of flow. In conclusion, the number of oscillations and extent of wave amplitude play vital role in transferring heat from a wavy surface. For the present collector design, augmentation in oscillations number as well as wave amplitude reduce the fluid domain but enlarge the surface area which is capable to collect more solar irradiation and consequently contribute to enhance heat transfer.

The effect of different wave amplitude on thermal field is presented in the middle columns of Figures 7 and 8. The isothermal lines are almost parallel to the nearest wall and the higher temperature gradient region remains in the vicinity of the concave portions of the upper wall which indicates the higher heat transfer region also because the space between the upper and lower surface is least on theses parts of the collector. When escalating the value of $A$, the isotherms slightly distorted from the convex portion of the upper surface and became denser near the concave portions where the temperature gradient become higher because these regions getting closer to the base wall making more heat transfer.

The third columns of Figures 7 and 8 display the concentration field of nanoparticle for different $A$. The overall appearance concerning the figures determines that thermophoresis and Brownian forces cause the nanoparticles concentration to grow nonuniform everywhere within the region. For any case, nanoparticle concentration remains higher at the top middle of each wave. The isoconcentrations are more concentrated near the walls with an increase of wave amplitude. The solutal boundary layer becomes thicker by increasing by $A$. Such performance designates the outline of the thermal carrier within nanofluid, and one requirement remains concerned concerning the near-wall section and hold lower or higher particle concentration which directs to higher or lower rates of heat transfer.

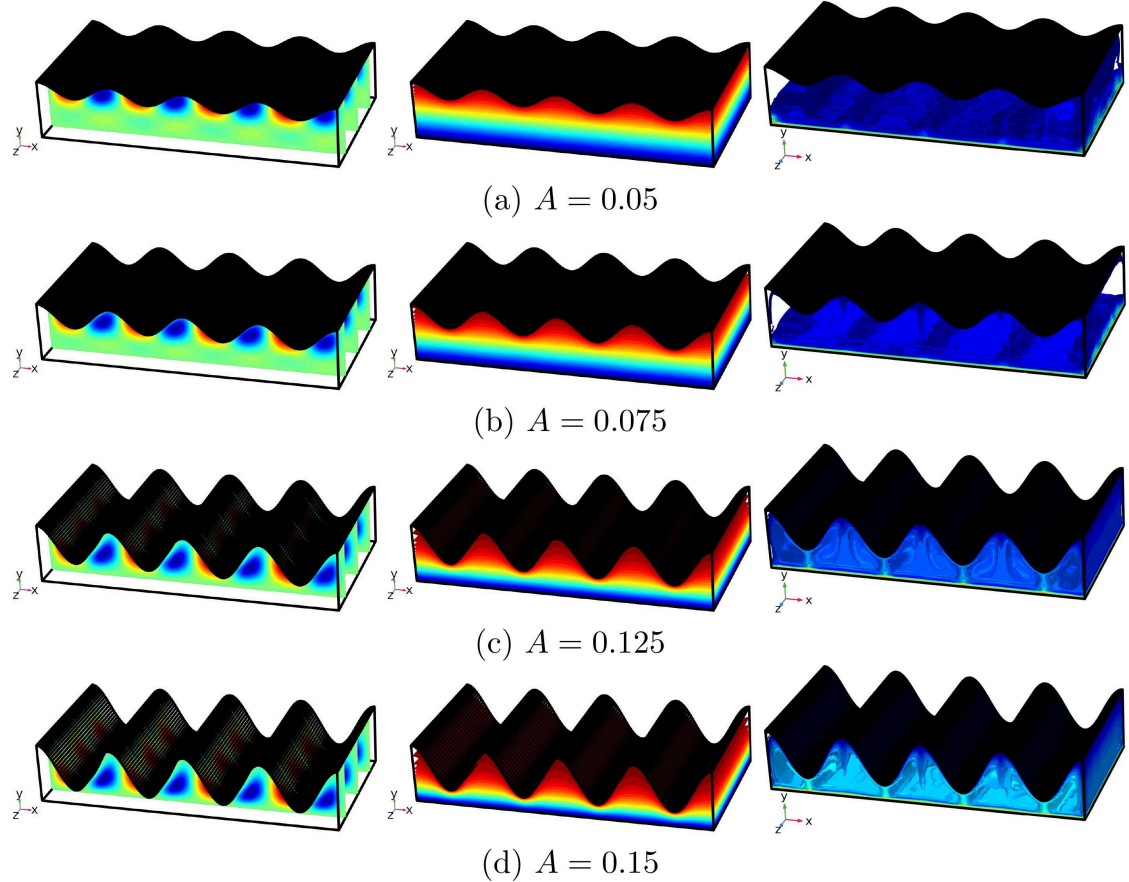

(a) $A = 0.05$

(b) $A = 0.075$

(c) $A = 0.125$

(d) $A = 0.15$

**Figure 7.** Variation of the 3D streamlines (**left**), isotherms (**middle**), and nanoparticle distribution (**right**) evolution by amplitude ($A$) for $N = 4$, $Ra = 10^6$, and $\phi = 0.02$.

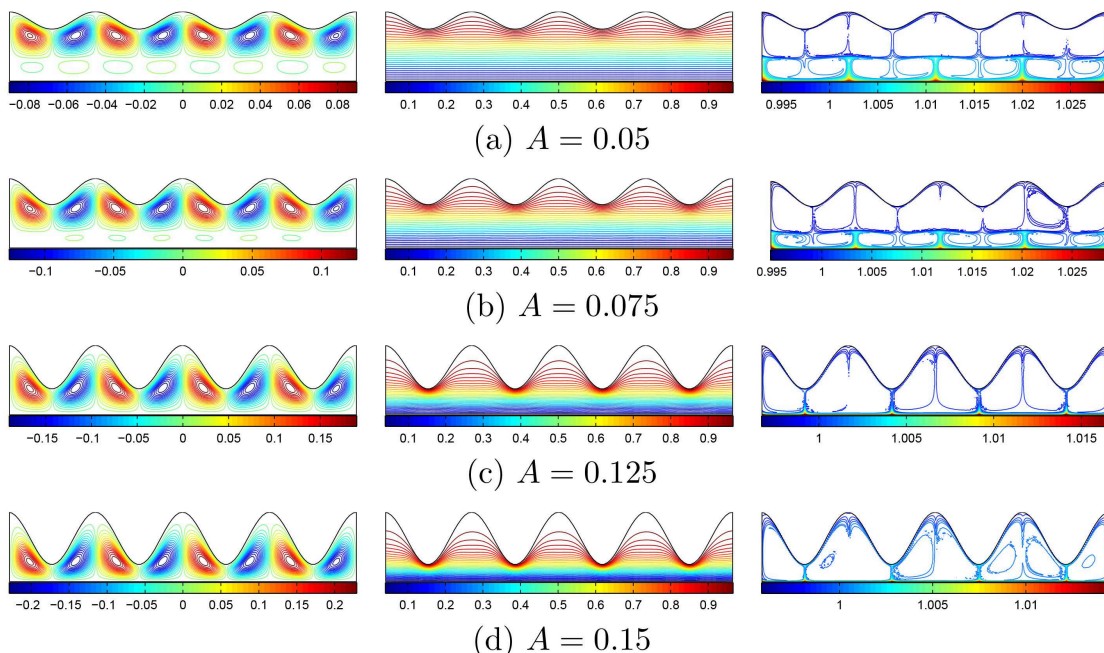

**Figure 8.** Variation of the streamlines (**left**), isotherms (**middle**), and nanoparticle distribution (**right**) evolution by amplitude ($A$) for $N = 4$, $Ra = 10^6$, and $\phi = 0.02$ in the plane $(X, Y)$.

Figure 9 shows the distribution of nanoparticles at the plane of $Y = 0.1$ along the $X$-direction. Figure 9a,b depicts the results for different $N$ and $A$, respectively. In both forms, a noticeable variation of particles concentration close to the horizontal surfaces can be recognized. A significant level of nanoparticles could be located at the lower surface. That is due to the thermophoresis force, which manages to move the nanoparticles of hot to cold zones. Close to the heated upper surface, the concentration of nanoparticles is low. This is again due to the thermophoresis effect, which sweeps the particles away from the hot surface. The intensity of the concentration boundary layer remains minimum compared toward the temperature and hydrodynamic boundary layers. This thin boundary layer is the result of the vast Lewis and Schmidt numbers for nanofluids. Distant from the surfaces, where the temperature gradients are smooth, a uniform concentration of nanoparticles could be found. The consistent level of nanoparticles at such region is due to the Brownian motion effects, which tend to move the nanoparticles from a high concentration area to a low concentration area. At the center of the collector, there is a small amount of nanoparticles, and therefore there is a low concentration gradient. In the case of high $N$ and $A$, the concentration profiles are much affected and shifted downward. This downward shift is due to the change of the cold flow toward the down bottom surface.

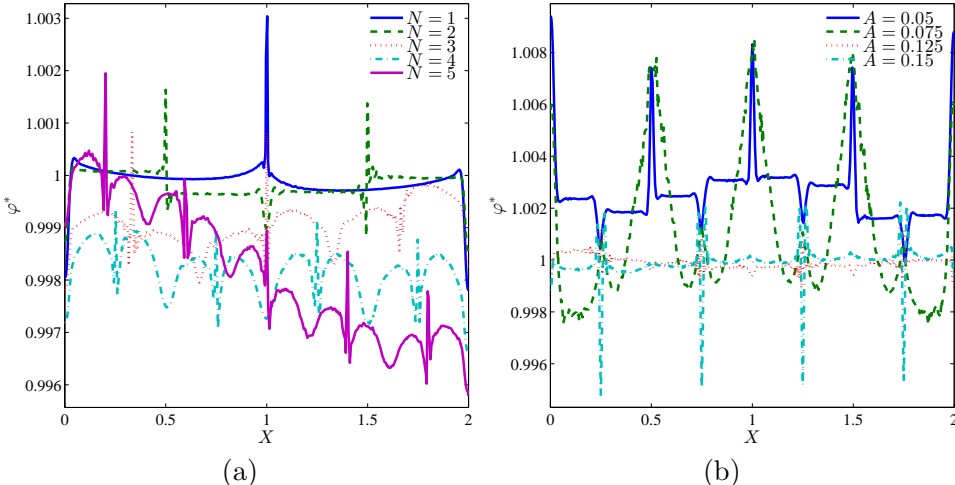

**Figure 9.** Variations of local normalized solid volume fraction interfaces with the horizontal line ($Y = 0.1$) for different (**a**) $N$ and (**b**) $A$ at $Ra = 10^6$ and $\phi = 0.02$.

### 4.3. Variation of the Nusselt Number

Figure 10 presents the local Nusselt number difference adjacent to the top heated wavy surface employing several $N$ and $A$ values. The overall evaluation shows that during both states, local maxima and minima values occur at the concave and the convex portion, respectively. An increment of the Nusselt number remains recognized concerning both cases when increasing the number of waves as well as wave amplitudes. This does observe notably that the heat transfer augmentation through examining the performance of wave amplitude is higher than the variation of the number of waves.

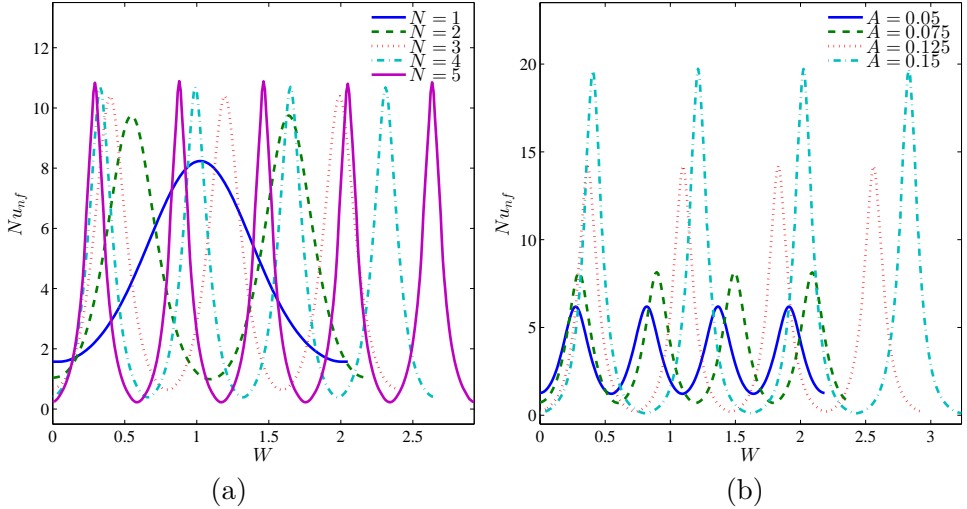

**Figure 10.** Variation of local Nusselt number interfaces with the wavy $W$ for different (**a**) $N$ and (**b**) $A$ at $Ra = 10^6$, and $\phi = 0.02$.

3-D and 2-D changes regarding average Nusselt number among *A* and *N* are given in Figure 11. As recognized by this figure, the average Nusselt number increases by boosting the values of both *A* and *N*. For both cases, the length of the wavy surface increased, leading to a greater energy shift of the hot surface toward the cold one. Heat transfer enhancement is more significant for the variation of *A* than that of *N*. Furthermore, average Nusselt number versus Rayleigh number during various *N* is also shown in Figure 11c. At lower *Ra* values, conductive heat transfer exists dominant than the convective heat transfer. Furthermore, developing the Rayleigh number leads to higher heat transfer due to the strong convection current.

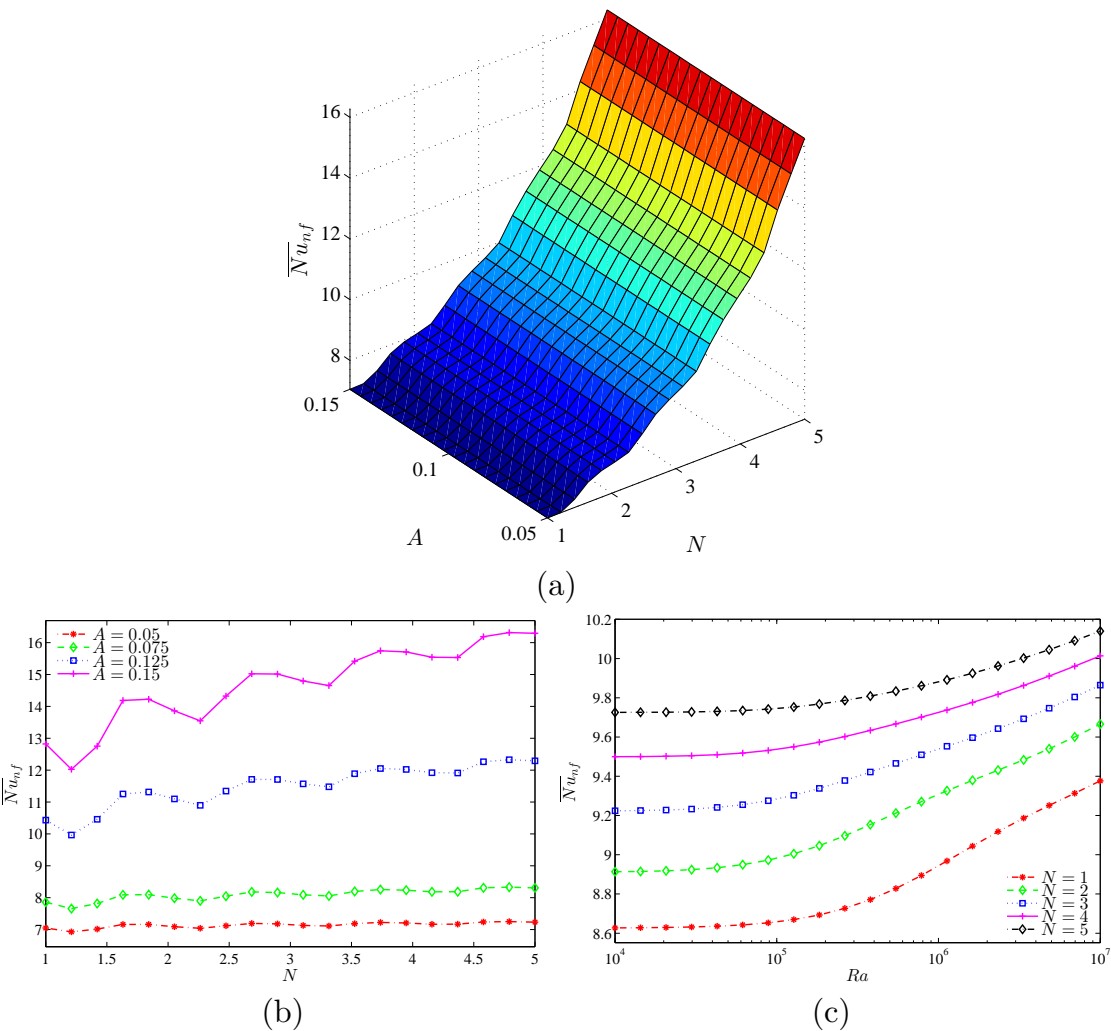

**Figure 11.** (**a**) 3D-plot of the average Nusselt number with *N* and *A*, (**b**) variation of the average Nusselt number with *N* for different *A*, and (**c**) variation of the average Nusselt number with *Ra* for different *N* at $\phi = 0.02$.

The variation in the average Nusselt number among volume fractions of nanoparticles of various *N* and *A* is shown in Figure 12a,b. The volume fraction of $\phi$ varies of 0 upon 0.04. According to this configuration, the average Nusselt number improves by the growing of $\phi$ toward both *N* and *A* variation. Causes higher thermal conductivity regarding nanoparticles, adding more particles improves the thermal energy transport within the two surfaces. More to the point, the variation from *A* and *N* has the similar effect as seen in Figure 10 concerning various volume fractions of nanoparticles. The outcome of *A* toward heat transfer occurs more noticeable than that of *N*.

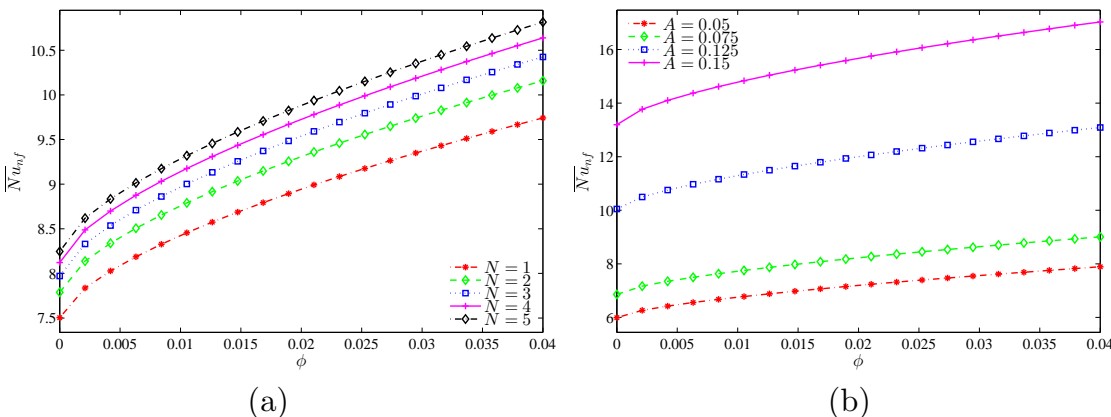

**Figure 12.** Variation of the average Nusselt number with $\phi$ for different (**a**) $N$ and (**b**) $A$ at $Ra = 10^6$.

### 4.4. Outcomes of Rayleigh Number (Ra)

Figure 13 illustrates the variety of the (left) streamlines, (middle) isotherms, and (right) nanoparticle concentration with Rayleigh number ($Ra$) for $N = 4$, $A = 0.1$, and $\phi = 0.02$ in the $X, Y$ plane. Rayleigh number holds a strong impact in the flow field. The vortices become stronger and more concentrated near the upper surface with the increasing $Ra$ values. At $Ra = 10^7$, some lower strength vortices appear close the cold bottom surface in addition to the strong vortices at the upper portion. The higher Rayleigh number indicates the higher temperature contrast within the hot and cold surfaces, which causes a stronger flow circulation inside the domain. Additionally, Rayleigh number associated with the buoyancy-driven flow. Increasing the Rayleigh number means the increase in the buoyancy forces and overcome the viscous forces. Therefore, convection currents dominate at high Rayleigh numbers. At high Rayleigh number, the temperature difference causes a higher density difference between the fluids near the cold and hot walls, leading to a strong convection current that improves the convective heat transfer rate accordingly.

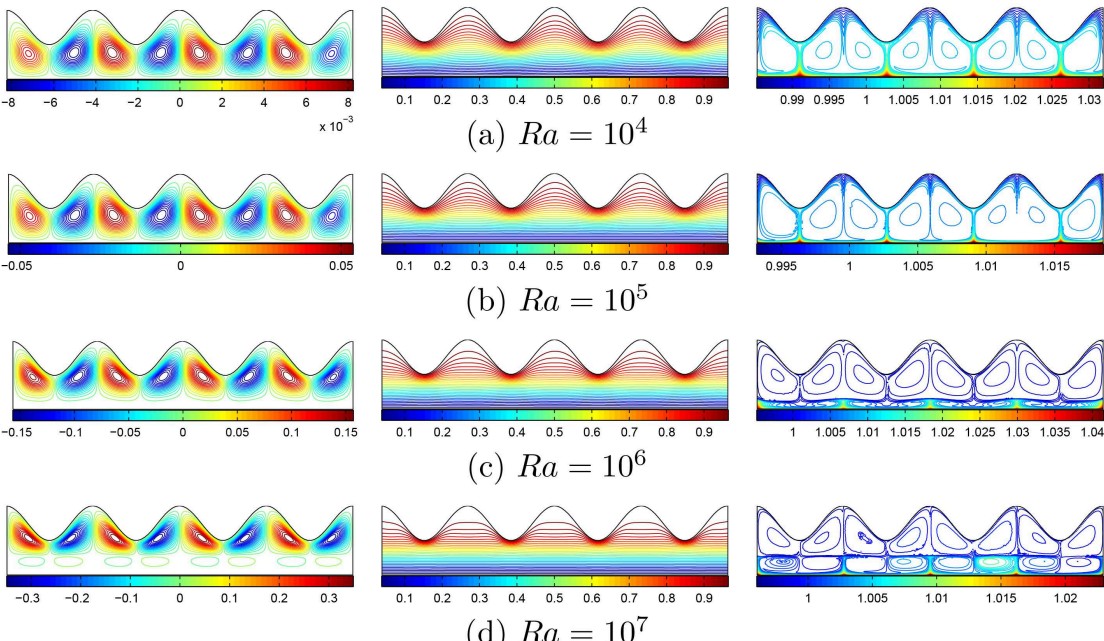

**Figure 13.** Variation of the streamlines (**left**), isotherms (**middle**), and nanoparticle distribution (**right**) evolution by Rayleigh number ($Ra$) for $N = 4$, $A = 0.1$, and $\phi = 0.02$ in the plane $(X, Y)$.

Looking at the middle column of Figure 13, it is observed that the isotherms are uniformly stratified throughout the whole collector. The isothermal lines move away from the upper wavy wall

as the value of *Ra* goes up. Due to increasing Rayleigh number, temperature gradient becomes greater, leading to more thermal energy transfer of the hot toward the cold surface into strong convection current. The locative concentration of nanoparticles from the upper and lower surfaces to the fluid for Rayleigh number variation is of great importance and is depicted in the right column of Figure 13. Nanoparticle distribution is more nonuniform for large Rayleigh numbers. Nevertheless, it remains further uniform toward low Rayleigh numbers. The concentration gradient near the top and bottom surface becomes more important for a higher values of *Ra*.

### 4.5. Outcomes of Solid Volume Fraction ($\phi$)

The simulation results toward the influence on the (left) streamlines, (middle) isotherms, and (right) nanoparticle concentration with solid volume fraction ($\phi$) at $N = 4$, $A = 0.1$, and $Ra = 10^6$ in the $X - Y$ plane is presented in Figure 14. This stands apparent of the streamlines the occurrence from the two circulations inside each wavy portion remains almost the same in size and strength. Aforementioned does assign into a particular buoyancy force ($Ra = 10^6$), adding nanoparticles up to 4% appears not to have an important influence toward the flow field. In this case, the density difference due to variable volume fraction concerning nanoparticles is suppressed by the buoyancy effect. More importantly, increasing solid volume fraction of nanoparticle up to a certain level (4%) is very effective in heat transfer enhancement. The physics behind this is that the nanofluid has higher thermal conductivity than the conventional fluid and heat transfer occurs at a higher rate in the high thermal conductivity medium. For adding more nanoparticles, the sedimentation process starts in the fluid and reduction of flow occurs which is not efficient in heat transfer in the desired rate.

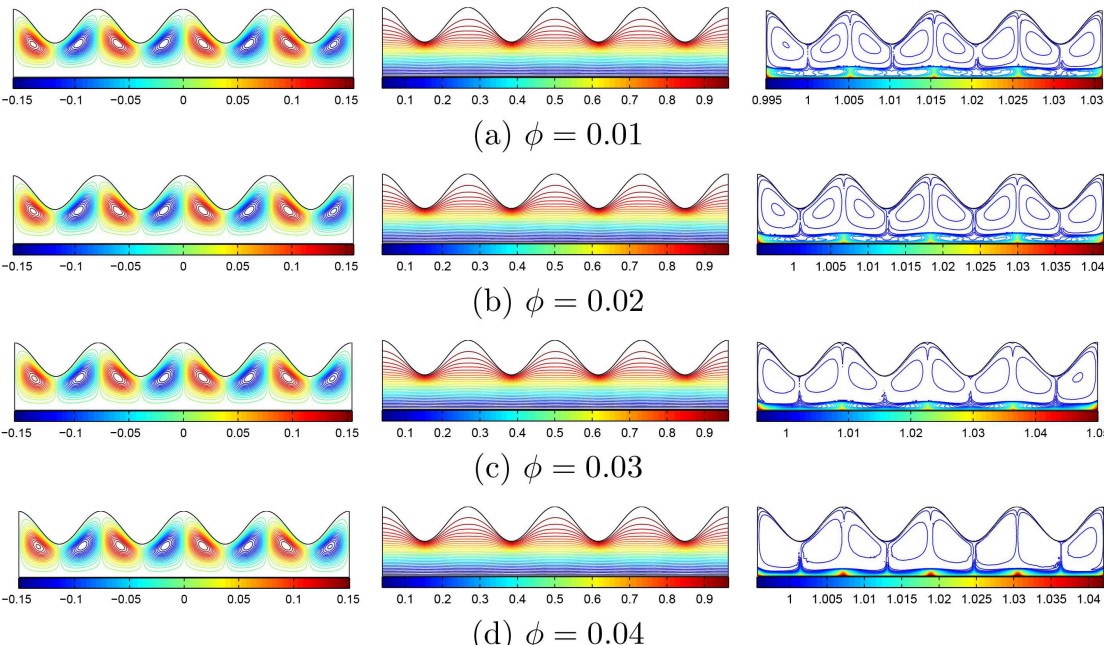

**Figure 14.** Variation of the streamlines (**left**), isotherms (**middle**), and nanoparticle distribution (**right**) evolution by solid volume fraction ($\phi$) for $N = 4$, $A = 0.1$, and $Ra = 10^6$ in the plane $(X, Y)$.

As shown in the second column of Figure 14, isothermal contours have no notable changes for the variation of solid volume fraction. Though the isothermal lines are nearly the same, increasing nanoparticle volume fraction increases the thermal conductivity regarding the nanofluid, which generates higher energy transport from heated to cooler portion, which is clear in Figure 12. From the last column of Figure 14, it is apparent that the nanoparticle volume fraction has a significant influence on the nanoparticle concentration field. At lower values of $\phi$, the concentration contours mostly remain near the walls indicating a higher concentration gradient. With the increasing $\phi$,

more lines appear in the domain. Thus, higher nanoparticle volume fraction reduces the concentration difference between the walls and the nanofluid. Consequently, less particle distribution will take place.

### 4.6. Variation of the Nusselt Number

Figure 15a describes the local Nusselt number variation adjacent to the top hot wavy surface with various values of $Ra$ at $N = 4$, $A = 0.1$, and $\phi = 0.02$. Local heat transfer appears among maximum and minimum rates toward the concave and convex portions of the collector. An increment of the buoyancy parameter ($Ra$) points toward an obvious enhancement on the local Nusselt number. It is affected by the high-temperature gradient and energy transporter that related to the high values of $Ra$. In addition, augmentation of nanoparticle volume fraction tends to boost the nanofluid thermal conductivity, and, in conclusion, an increment of the heat transfer does obtain. Such a behavior is caused higher thermal gradient and energy transport with the rising concentration of nanoparticle volume fraction, as specified in Figure 15b.

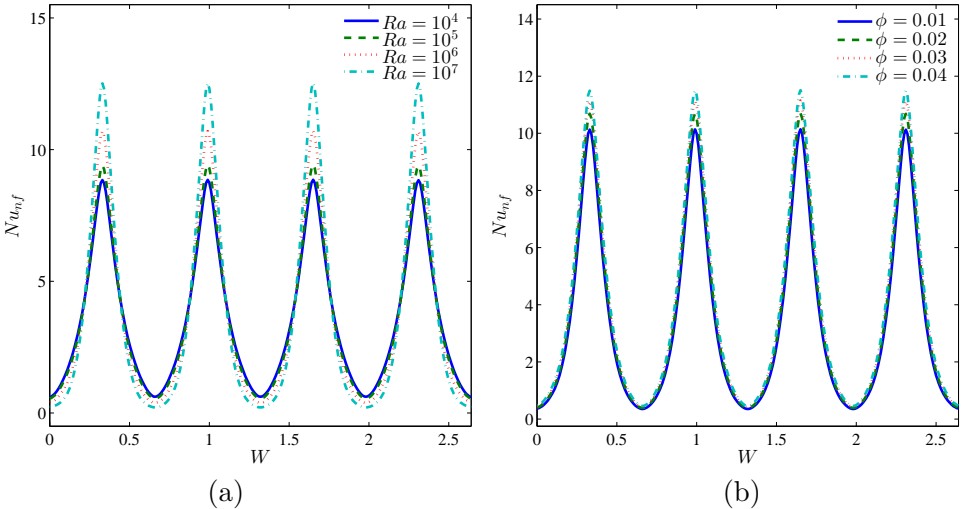

**Figure 15.** Variation of local Nusselt number interfaces with the wavy $W$ for different (**a**) $Ra$ and (**b**) $\phi$ at $N = 4$ and $A = 0.1$.

Figure 16 explains the 3-D and 2-D changes of average Nusselt number among $Ra$ and $\phi$ at $A = 0.1$. As clearly observed from this figure that the convective heat transfer augments with the increment of both Rayleigh number and nanoparticle volume fraction. Increasing $Ra$ points through the convective heat transfer, which shows higher thermal gradient with the following of $\phi$ increasing. More importantly, an evident enhancement is recorded toward the values concerning the average Nusselt number, including raising $N$ with respect to all values of $\phi$.

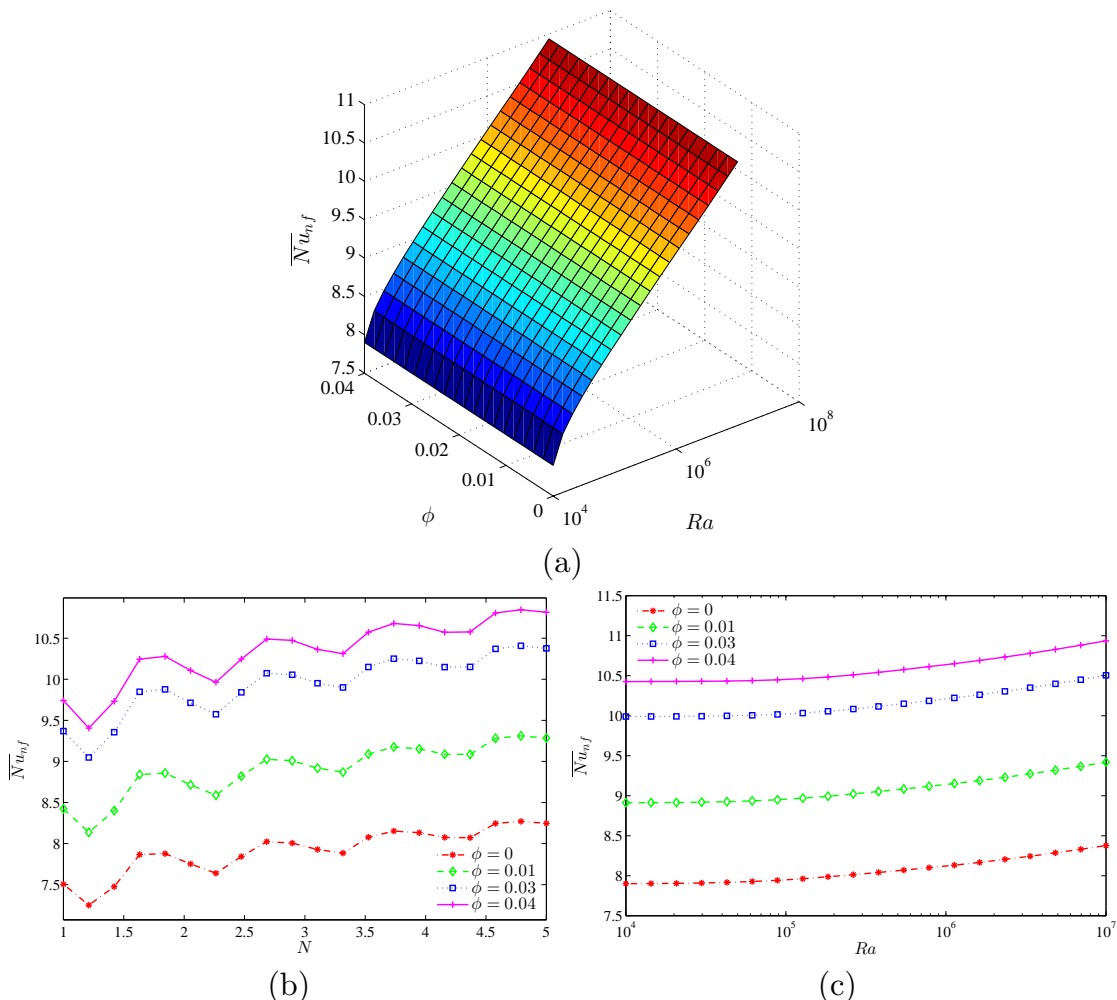

**Figure 16.** (**a**) 3D-plot of the average Nusselt number with *Ra* and $\phi$, (**b**) variation of the average Nusselt number with *N* for different $\phi$, and (**c**) variation of the average Nusselt number with *Ra* for different $\phi$ at $N = 4$ and $A = 0.1$.

## 5. Conclusions

In this numerical attempt, the two-phase (non-homogeneous) nanofluid pattern concerning convection heat transfer within a 3D wavy direct absorber solar collector is examined. The governing equations of the Navier–Stokes including the energy equations of the nanofluid are transformed into dimensionless form and then solved numerically by employing the Galerkin weighted residual finite-element technique. Impacts of various parameters such as the number of oscillations, wave amplitude, Rayleigh number, and nanoparticle volume fraction on the streamlines, isotherms, nanoparticle concentration, and heat transfer are obtained and discussed in this work. The following concluding remarks are drawn from the current analysis.

1. Increasing the number of oscillations leads to a higher temperature gradient region which is extended to the five concave portions of the top wall due to the extra energy of the upper wall.
2. It is found that an augmentation of the wave amplitude enhances the thermophoresis and Brownian influences, which clearly force the nanoparticles concentration to display a completely nonuniform trend in the region.
3. An augmentation in the local heat transfer is observed for both cases when increasing the number of waves as well as the wave amplitudes. However, this augmentation in the heat transfer is noticed to be higher by considering the role of the wave amplitude.

4. At lower Rayleigh number values, conductive heat transfer is more dominant than the convective heat transfer. Moreover, improving the Rayleigh number leads to higher heat transfer due to the strong convection current.
5. The average Nusselt number develops upon raising both the wave amplitude and the number of oscillations. For both cases, growing the length of the wavy surface results in more energy transfer from the hot surface. More importantly, the heat transfer enhancement is observed more significantly with the variation of the wave amplitude.
6. Augmentation of the nanoparticles concentration tends to boost the nanofluid thermal conductivity, and as a result, a gain in the average Nusselt number is obtained due to the higher thermal gradient and energy transport.

**Author Contributions:** Conceptualization, A.I.A., S.P., M.G., A.J.C., and I.H.; Methodology, A.I.A., S.P., M.G., and A.J.C.; Software, A.I.A., M.G., and A.J.C.; Validation, A.I.A., S.P., M.G., A.J.C., and I.H.; Investigation, A.I.A., S.P., M.G., A.J.C., and I.H.; Writing—Original Draft Preparation, A.I.A., S.P., M.G., A.J.C., and I.H.; Writing—Review and Editing, A.I.A., S.P., M.G., A.J.C., and I.H.; Funding Acquisition, I.H. All authors have read and agreed to the published version of the manuscript.

**Funding:** This research was funded by Universiti Kebangsaan Malaysia (UKM) grant number GP-2019-K006388.

**Conflicts of Interest:** The authors declare no conflicts of interest.

## Abbreviations

The following abbreviations are used in this manuscript.

Nomenclature

| | |
|---|---|
| $A$ | amplitude |
| $C_p$ | specific heat capacity |
| $d_f$ | diameter of the base fluid molecule |
| $d_p$ | diameter of the nanoparticle |
| $D_B$ | Brownian diffusion coefficient |
| $D_{B0}$ | reference Brownian diffusion coefficient |
| $D_T$ | thermophoretic diffusivity coefficient |
| $D_{T0}$ | reference thermophoretic diffusion coefficient |
| $H$ | thickness of solar collector |
| $k$ | thermal conductivity |
| $L$ | length of the solar collector |
| $Le$ | Lewis number |
| $N$ | number of oscillations |
| $N_{BT}$ | ratio of Brownian to thermophoretic diffusivity |
| $\overline{Nu}$ | average Nusselt number |
| $Pr$ | Prandtl number |
| $Ra$ | Rayleigh number |
| $Re_B$ | Brownian motion Reynolds number |
| $S$ | total length of the wavy heater |
| $Sc$ | Schmidt number |
| $T$ | temperature |
| $T_0$ | reference temperature (310K) |
| $T_{fr}$ | freezing point of the base fluid (273.15K) |
| **v** | velocity vector |
| **V** | normalized velocity vector |
| $u_B$ | Brownian velocity of the nanoparticle |
| $W$ | width of the solar collector |
| $x, y, z$ & $X, Y, Z$ | space coordinates & dimensionless space coordinates |

Greek symbols

| | |
|---|---|
| $\alpha$ | thermal diffusivity |
| $\gamma$ | inclination angle of magnetic field |
| $\beta$ | thermal expansion coefficient |
| $\delta$ | normalized temperature parameter |
| $\theta$ | dimensionless temperature |
| $\mu$ | dynamic viscosity |
| $\nu$ | kinematic viscosity |
| $\rho$ | density |
| $\varphi$ | solid volume fraction |
| $\varphi^*$ | normalized solid volume fraction |
| $\phi$ | average solid volume fraction |

Subscript

| | |
|---|---|
| $c$ | cold |
| $f$ | base fluid |
| $h$ | hot |
| $nf$ | nanofluid |
| $p$ | solid nanoparticles |

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
