# Peer review of "Convection Heat Transfer in 3D Wavy Direct Absorber Solar Collector Based on Two-Phase Nanofluid Approach"

_applsci, doi:10.3390/app10207265_

Round 1

Reviewer 1 Report

The paper "Convection heat transfer in 3D wavy direct absorber solar collector based on two-phase nanofluid approach" deals with an interesting subject, but it needs some revisions:

1) The authors should carefully review the manuscript for the language. In some parts the paper is not clear.

2) A nomenclature should be added in order to list all the symbols and the corresponding units of measure.

3) A deeper bibliographic analysis is required in order to give a complete understanding of the state of the art. It is suggested to add these papers (and more):
Optical absorption measurements of oxide nanoparticles for application as nanofluid in direct absorption solar power systems - Part I: Water-based nanofluids behavior, (2016) Solar Energy Materials and Solar Cells;
Optical absorption measurements of oxide nanoparticles for application as nanofluid in direct absorption solar power systems - Part II: ZnO, CeO2, Fe2O3 nanoparticles behavior, (2016) Solar Energy Materials and Solar Cells;

An investigation of layering phenomenon at the liquid-solid interface in Cu and CuO based nanofluids", (2016) International Journal of Heat and Mass Transfer, Volume 103, pp. 564-571;
Measurement and control system for thermosolar plant and performance comparison between traditional and nanofluid solar thermal collectors, (2016) International Journal on Smart Sensing and Intelligent Systems, vol. 9, no. 3, pp. 681 - 708;
An explanation of the Al2O3 nanofluid thermal conductivity based on the phonon theory of liquid", (2016) Energy, 116, pp. 786-794
Numerical simulation of thermal efficiency of an innovative Al2O3 nanofluid solar thermal collector: Influence of nanoparticles concentration, 2016, Thermal Science, Volume 2016, 12p, DOI: 10.2298/TSCI151207168C

Experimental investigation of transparent parabolic trough collector based on gas-phase nanofluid, Applied Energy, Volume 203, 1 October 2017, Pages 560-570

A critical analysis of clustering phenomenon in Al2O3 nanofluids, 2019, Journal of Thermal Analysis and Calorimetry 135(1), pp. 371-377, DOI: 10.1007/s10973-018-7099-9

4) The experimental validation is very limited and it is not general. There is not an error analysis of the results.

5) More details should be given on how the thermophysical properties of the nanofluids are accounted for (there is only a table with properties at only one temperature, there are not data about temperature dependance).

6) More comments explaining the physical effects which determined the obtained results are required. Only descriptions of the plots are presented, with limited physical explanation.

7) The authors could group the plots in less images, in order to reduce the number of pictures
8) It is not clear the advantage of the mathematical model and its scientific relevance due to its intrinsic limitations.

Author Response

First of all, we thank the respected referees for their constructive comments which clearly enhanced the quality of the manuscript. Our replies to the comments are given below:

Reviewer #1

Comment 1. The authors should carefully review the manuscript for the language. In some parts
the paper is not clear.
Reply: We have carefully proofread the revised manuscript and corrected the unclear
parts in the the revised manuscript.
Comment 2. A nomenclature should be added in order to list all the symbols and the corresponding
units of measure.
Reply: We have included the Nomenclature section just before the References section
of the revised version.
Comment 3. A deeper bibliographic analysis is required in order to give a complete understanding
of the state of the art. It is suggested to add these papers (and more):
Optical absorption measurements of oxide nanoparticles for application as nanofluid
in direct absorption solar power systems - Part I: Water-based nanofluids behavior,
(2016) Solar Energy Materials and Solar Cells; Optical absorption measurements
of oxide nanoparticles for application as nanofluid in direct absorption solar
power systems - Part II: ZnO, CeO2, Fe2O3 nanoparticles behavior, (2016) Solar
Energy Materials and Solar Cells; An investigation of layering phenomenon at the
liquid-solid interface in Cu and CuO based nanofluids”, (2016) International Journal
of Heat and Mass Transfer, Volume 103, pp. 564-571; Measurement and control
system for thermosolar plant and performance comparison between traditional and
nanofluid solar thermal collectors, (2016) International Journal on Smart Sensing
and Intelligent Systems, vol. 9, no. 3, pp. 681 - 708;
An explanation of the Al2O3 nanofluid thermal conductivity based on the phonon
theory of liquid”, (2016) Energy, 116, pp. 786-794
Numerical simulation of thermal efficiency of an innovative Al2O3 nanofluid solar
thermal collector: Influence of nanoparticles concentration, 2016, Thermal Science,
Volume 2016, 12p, DOI: 10.2298/TSCI151207168C
Experimental investigation of transparent parabolic trough collector based on gasphase
nanofluid, Applied Energy, Volume 203, 1 October 2017, Pages 560-570
A critical analysis of clustering phenomenon in Al2O3 nanofluids, 2019, Journal
of Thermal Analysis and Calorimetry 135(1), pp. 371-377, DOI: 10.1007/s10973-
018-7099-9
Reply: We have updated the literature review to contain the mentioned articles. Some
descriptions of the types of works done by these authors are also included.
Comment 4. The experimental validation is very limited and it is not general. There is not an
error analysis of the results.
Reply: We have updated the validation part and included more validations with experimental
data including an error analysis. Please see the new Fig. 4 in the revised
version.
Comment 5. More details should be given on how the thermophysical properties of the nanofluids
are accounted for (there is only a table with properties at only one temperature,
there are not data about temperature dependance).
Reply: Many thanks for the comment form the respected reviewer. However, in the
present work, the properties of nanofluid are considered as constant with respect to
temperature. In our future work, temperature-dependent properties like the thermal
conductivity and the viscosity will be analyzed.
Comment 6. More comments explaining the physical effects which determined the obtained results
are required. Only descriptions of the plots are presented, with limited physical
explanation.
Reply:We have included more explanations regarding to the the physical effects. Please
see the highlighted parts in the Results and Discussion section.
Comment 7. The authors could group the plots in less images, in order to reduce the number of
pictures.
Reply: We have already included number of figures in one single image.
Comment 8. It is not clear the advantage of the mathematical model and its scientific relevance
due to its intrinsic limitations.
Reply: The reason behind selecting the present mathematical model is that the twophase
mixture method of Buongiorno [Ref. 1] used in this study provides more
accurate results compared to the single-phase methods. Other studies [Ref. 2, Ref.
3] also confirm the better accuracy of the model of Buongiorno with respect to experimental
measurements.
Ref. 1. J. Buongiorno, Convective transport in nanofluids, J. Heat Transf. 128 (2006)
240250.
Ref. 2. H.A. Pakravan,M. Yaghoubi, Analysis of nanoparticlesmigration on natural
convective heat transfer of nanofluids, Int. J. Therm. Sci. 68 (2013) 7993.
Ref. 3. G.A. Sheikhzadeh, M. Dastmalchi, H. Khorasanizadeh, Effects of nanoparticles
transport mechanisms on Al2O3water nanofluid natural convection in a square
enclosure, Int. J. Therm. Sci. 66 (2013) 5162.

Thank you for your support.

Sincerely yours,

Reviewer 2 Report

The submitted manuscript presents numerical attempt of the two-phase (non-homogeneous) nanofluid approach toward the convection heat transfer within 3D wavy direct absorber solar collector. The manuscript is in general well organized and it would be of interest to the research community in this field of work as it offers results that are of not only theoretical value, but also useful in practice. For instance, the review and the drawn conclusions would support developers and thermal engineers in their work.

In order to improve the readability and clarity of the manuscript, a few minor remarks need to be addressed before the paper is to be accepted for publishing:

1) All the references of equations in the text are missing! Please complete the text with Equation (1)…(26) as references!

2) The font size on more figures (especially on Figures 4; 9; 10; 11; 16) should be increased for better readability!

3) The introduction section could be extended more. There are many other research results in this field that should be mentioned by the authors as research backgrounds in the “Introduction” section of the paper. By this way the “Introduction” and also the “References” sections of this paper should be completed with the under mention relevant references especially that relates to this field:

[7] Xu, B., Xu, J., Chen, Z. Heat transfer study in solar collector with energy storage. International Journal of Heat and Mass Transfer. International Journal of Heat and Mass Transfer 2020, 156, 119778.

[8] Behura, A.K., Gupta, H.K. Efficient Direct Absorption Solar Collector Using Nanomaterial Suspended Heat Transfer Fluid. Materials Today: Proceedings 2020, 22(4), 1664-1668.

Please put these references into the text of the first sentence (in the Introduction section):

Solar thermal collector is a device which utilizes the solar energy via collecting and concentrating solar radiation [7-8].”

Please complete the References section of your paper with these referred papers: with numbers [7]; [8] references!

4) The reference style of the “References“ section of the recent paper does not meet with the requirements of the journal. Please check the relating formal requirements in the “guide for authors” again and correct it following the instructions!

To improve the paper based on these minor modifications are very significant to have success in acceptance for publication!

Thank you for your consideration in advance!

Author Response

First of all, we thank the respected referees for their constructive comments which clearly enhanced the quality of the manuscript. Our replies to the comments are given below:

Reviewer #2

Comment 1. All the references of equations in the text are missing! Please complete the text with Equation (1)(26) as references!
Reply: We have cited appropriate reference to Eqs. (1)–(26) for underived equations.
Comment 2. The font size on more figures (especially on Figures 4; 9; 10; 11; 16) should be
increased for better readability!
Reply: We have tried our best to increased the font size in all figures of the revised
version.
Comment 3. The introduction section could be extended more. There are many other research results
in this field that should be mentioned by the authors as research backgrounds
in the Introduction section of the paper. By this way the Introduction and also the
References sections of this paper should be completed with the under mention relevant
references especially that relates to this field:
- Xu, B., Xu, J., Chen, Z. Heat transfer study in solar collector with energy storage.
International Journal of Heat and Mass Transfer. International Journal of Heat and
Mass Transfer 2020, 156, 119778.
- Behura, A.K., Gupta, H.K. Efficient Direct Absorption Solar Collector Using
Nanomaterial Suspended Heat Transfer Fluid. Materials Today: Proceedings 2020,
22(4), 1664-1668.
Please put these references into the text of the first sentence (in the Introduction
section):
Solar thermal collector is a device which utilizes the solar energy via collecting and
concentrating solar radiation [7-8].
Please complete the References section of your paper with these referred papers:
with numbers [7]; [8] references!
Reply: We have updated the literature review to contain the mentioned articles. Some
descriptions of the types of works done by these authors are also included.
Comment 4. The reference style of the References section of the recent paper does not meet with
the requirements of the journal. Please check the relating formal requirements in the
guide for authors again and correct it following the instructions!
Reply: The reference list has been unified according to the journal style.

Thank you for your support.

Sincerely yours,

Round 2

Reviewer 1 Report

The revision is ok